# Breaking down linguistic barriers: The radical impact of translanguaging on pre-service EFL teachers' perspectives in Turkey

Ömer Gökhan Ulum  *

Department of English Language Education, Education Faculty, Mersin University, Yenişehir, Turkey

* gokhanulum@mersin.edu.tr, omergokhanulum@gmail.com

## Abstract

This study examines the perceptions and practices of translanguaging among pre-service English as a Foreign Language (EFL) teachers in Turkey, employing a mixed-method research design with a phenomenological framework. The sample consists of 401 pre-service EFL teachers from the Education Faculty's English Language Teaching (ELT) Department at a state university during the 2024–2025 academic year. Data collection included a rigorously developed 28-item Likert-scale questionnaire and semi-structured interviews. The questionnaire, refined through factor analysis, explored various dimensions of translanguaging, including its perceived benefits, challenges, and usage contexts. Quantitative data were analyzed using descriptive and inferential statistics, while qualitative data were examined through thematic analysis. The results suggest that translanguaging is widely perceived as a valuable and natural pedagogical strategy, as it enhances learning by reducing language anxiety, fostering inclusivity, and promoting active participation. Its "natural" aspect lies in the alignment with learners' spontaneous use of their full linguistic repertoire, facilitating smoother integration into the learning process. Translanguaging is shown to alleviate anxiety, build confidence, and stimulate classroom engagement. This research underscores the significance of integrating translanguaging in teacher education programs and highlights the need for further investigation into its diverse effects on language learning and teaching practices.

## 1. Introduction

In recent years, the concept of translanguaging has garnered significant attention in the field of language education [1–6]. Translanguaging, a strategy involving multiple languages within a classroom setting, allows students and teachers to draw on their full linguistic repertoire during lessons [1, 7, 8]. This approach recognizes and utilizes the languages students are already familiar with [9], such as Turkish, Azerbaijani, Kurdish, Arabic, and English, throughout various stages of instruction. The strategy of translanguaging in English lessons can be seen from multiple perspectives [10]. Leung and Valdés [4] argue that it is a natural and beneficial practice that supports language learning by reducing anxiety [11], increasing confidence [12], and

**Competing interests:** The authors have declared that no competing interests exist.

facilitating comprehension [13]. On the other hand, translanguaging is also viewed as a consequence of inadequate English language proficiency, whether in terms of linguistic skills, speaking abilities, or confidence in using English exclusively [14]. The primary aim of this research is to explore the perceptions of pre-service English teachers regarding the use of translanguaging in English lessons. Specifically, the study will investigate whether translanguaging is perceived as a useful pedagogical tool or a necessity born out of linguistic deficiencies. Furthermore, the research will examine the potential benefits of translanguaging for enhancing participation, motivation, effective communication, creative thinking, and critical reflection among students.

The following research questions guide this investigation:

Main Research Question

How do various factors influence translanguaging among pre-service English as a Foreign Language (EFL) teachers in state university settings?

Sub-Research Questions

1. To what extent do gender differences impact the utilization of translanguaging in EFL environments?

2. How does the application of translanguaging vary across different classes within EFL settings?

3. What are the general perceptions of pre-service EFL teachers at a state university regarding the translanguaging strategy in EFL contexts?

4. How do pre-service EFL teachers at a state university perceive the application of translanguaging within specific EFL contexts?

5. What are the perceived benefits of translanguaging according to pre-service EFL teachers at a state university?

## 2. Background of the study

The concept of translanguaging, initially developed within the context of bilingual education, has its origins in the work of Welsh scholars during the 1980s. Colin Baker is credited with coining the term translanguaging when he encountered the work of Cen Williams, a leader in Welsh language revitalization programs [15–17]. Originally applied to Welsh-English bilingual classrooms, translanguaging referred to the systematic use of both languages for different instructional purposes [18]. The practice allowed students to engage in a dynamic interchange between languages, with the aim of promoting deeper understanding and greater cognitive engagement through both languages. As translanguaging evolved, it began to challenge the traditional notion of language separation, which had long dominated bilingual education [19]. Instead of viewing languages as distinct and isolated entities, translanguaging emphasized the fluidity of language use, where speakers drew from their entire linguistic repertoire to communicate effectively. This shift in perspective led to the expansion of translanguaging beyond Welsh-English contexts, influencing multilingual classrooms globally [14]. In the late 1990s and early 2000s, translanguaging was further conceptualized by researchers such as García [20] and Otheguy, García, and Reid [21], who developed it into a robust sociolinguistic theory. This theory positioned translanguaging as a practice that not only facilitates communication but also serves as a form of resistance to monolingual ideologies, which often marginalize multilingual speakers. By recognizing the legitimacy of all linguistic resources, translanguaging has

become a powerful pedagogical tool in advocating for language rights and equity in education. Furthermore, scholars like Canagarajah [22, 23] expanded translanguaging as an innovative teaching strategy, wherein both students and teachers strategically use multiple languages to foster deeper learning, encourage critical thinking, and bridge gaps between languages and cultures. This pedagogical shift reflects a growing awareness of the sociocultural and cognitive benefits of integrating multiple languages in the classroom, rather than enforcing strict language boundaries.

The history of translanguaging demonstrates its development from a practical tool in bilingual education to a comprehensive framework for understanding and utilizing multilingualism in educational settings. As a result, translanguaging is now widely recognized as a critical strategy for promoting inclusivity and engagement in diverse learning environments [15, 16, 24, 25]. Its application extends beyond language instruction, influencing policies and practices aimed at empowering multilingual students and fostering culturally responsive pedagogy.

Translanguaging, a concept rooted in bilingual education [1], has evolved as a critical pedagogical strategy in multilingual classrooms [26]. Originating from the work of Welsh scholars in the 1980s, translanguaging involves the fluid and dynamic use of multiple languages to facilitate learning and communication [15, 16, 18]. It challenges the traditional view of language separation [27, 28] and promotes the integration of students' linguistic resources in a cohesive and meaningful manner [29, 30]. In the context of English language teaching, translanguaging has gained prominence as educators recognize the diverse linguistic backgrounds of their students [31, 32]. Turkey, characterized by its rich linguistic diversity, presents a unique landscape for exploring the implications of translanguaging in English language classrooms [33, 34]. For students who speak Turkish, Azerbaijani, Kurdish, and Arabic, English language instruction in Turkey provides a fertile ground for implementing and studying translanguaging practices [35]. The theoretical foundations of translanguaging are supported by several educational theories, including Vygotsky's sociocultural theory, which emphasizes the role of social interaction and cultural tools in learning [36, 37], and Cummins' threshold hypothesis, which suggests that a certain level of proficiency in both languages is necessary for cognitive benefits to manifest [38, 39]. Translanguaging aligns with these theories by leveraging students' existing linguistic competencies to enhance their learning experiences. Empirical studies on translanguaging have demonstrated its potential benefits in various educational settings [1, 6, 40]. For instance, research has shown that translanguaging can reduce language anxiety [41], increase student participation [42], and foster a deeper understanding of the subject matter [43]. It also allows students to express complex ideas and engage in higher-order thinking using their full linguistic repertoire [44]. Additionally, translanguaging has been found to support the maintenance and development of students' home languages, contributing to their linguistic and cultural identity [29, 40, 45]. Despite its advantages, translanguaging presents challenges and raises concerns [46, 47]. Some educators and stakeholders worry that it might hinder the development of English proficiency, particularly if students rely too heavily on their first languages [48]. There are also concerns about the feasibility of implementing translanguaging practices in classrooms with varying levels of language proficiency among students [3].

In Turkey, the adoption of translanguaging in English language classrooms is influenced by several factors [34, 49, 50], including teachers' beliefs [50, 51], instructional practices [52], and institutional policies [50, 52]. Teachers' perceptions of translanguaging play a crucial role in its implementation [50, 53]. While some teachers view it as a valuable strategy to support learning [10, 50], others perceive it as a sign of linguistic deficiency or lack of confidence in using English [48, 54].

This study seeks to investigate these diverse perspectives and contribute to the ongoing discourse on translanguaging in English language education. By examining the perceptions of

both teachers and students, the research aims to provide a comprehensive understanding of the role of translanguaging in English language classrooms in Turkey. The findings will offer insights into how translanguaging can be effectively integrated into language teaching practices, addressing its potential benefits and challenges. Ultimately, this study aims to inform educators, policymakers, and researchers about the practical implications of translanguaging, promoting its thoughtful and strategic use to enhance language learning outcomes and support the linguistic and cultural diversity of students in Turkey.

## 2.1 Significance of the study

The significance of this study on translanguaging in English language classrooms in Turkey is multifaceted, addressing several critical areas in language education and broader educational practices. First and foremost, this study contributes to understanding how translanguaging can enhance language learning. By leveraging students' full linguistic repertoire, translanguaging can deepen comprehension, facilitate learning complex concepts, and promote critical thinking. This research will provide empirical evidence on the effectiveness of translanguaging in improving English proficiency, thus offering valuable insights for language educators seeking to adopt more inclusive and effective teaching strategies. In a country as linguistically diverse as Turkey, promoting multilingualism is essential. This study highlights the importance of maintaining and valuing students' home languages while learning English. The findings will underscore the role of translanguaging in supporting linguistic diversity, helping educators foster an environment where multiple languages are seen as assets rather than barriers. This approach aligns with global educational trends that emphasize the value of multilingualism in fostering cognitive, social, and cultural development. Teacher perceptions and practices play a crucial role in successfully implementing translanguaging. By exploring the attitudes and beliefs of English teachers towards translanguaging, this study will identify the factors that influence their willingness and ability to integrate this strategy into their teaching. Understanding these factors will enable the development of targeted professional development programs and resources that support teachers in effectively utilizing translanguaging to enhance their instructional practices. Language anxiety is a significant barrier to language acquisition [55–59]. This study will examine how translanguaging can reduce anxiety and increase students' confidence in using English. By providing a supportive environment where students can draw on their linguistic strengths, translanguaging can help mitigate the stress associated with learning a new language. The insights from this research will be valuable for educators and policymakers aiming to create more inclusive and supportive language learning environments. The findings of this study have important implications for educational policy. Policymakers will better understand the benefits and challenges of translanguaging, informing decisions related to language education curricula and instructional strategies. By highlighting the practical benefits of translanguaging, this study can advocate for policies that support multilingual education and recognize the linguistic rights of students. Finally, this study significantly contributes to the academic literature on translanguaging and multilingual education. It comprehensively analyses the perceptions and practices surrounding translanguaging in a specific context, enriching the global discourse on language education. The research findings will serve as a valuable resource for scholars, educators, and students interested in the intersections of language, culture, and education. In summary, this study is significant because it explores the potential of translanguaging to enhance language learning, support multilingualism, inform teacher practices, reduce language anxiety, and shape educational policies. By addressing these areas, the research aims to promote a more inclusive and effective approach to language education that values and leverages the linguistic diversity of students in Turkey.

## 3. Methodology

### 3.1 Research design

This study employs a mixed-method research design, integrating both quantitative and qualitative approaches to comprehensively understand the perceptions and practices surrounding translanguaging in English language classrooms. Specifically, the research follows a phenomenological approach to explore the lived experiences and perspectives of pre-service English as a Foreign Language (EFL) teachers at a state university in Turkey.

### 3.2 Participants

This study's participants are 401 pre-service EFL teachers enrolled in the Education Faculty, English Language Teaching (ELT) Department at a state university in Turkey for the 2024–2025 academic year. The participants include 45 preparatory class students, 49 first-year students, 129 second-year students, 100 third-year students, and 78 fourth-year students. The sample comprises 243 female and 158 male participants, representing the student body differently. The present study has been reviewed by the Mersin University Social and Human Sciences Ethics Committee. The study was found to be ethically appropriate in the decision dated July 11, 2024, with Decision Number 249. The recruitment period for the data collection were between July 11–20, 2024. Additionally, written informed consent was provided by all participants. Participants were assured of the confidentiality and anonymity of their responses, and they were informed of their right to withdraw from the study at any time without any consequences.

### 3.3 Data collection instruments

**3.3.1 Questionnaire.** A questionnaire was developed to collect quantitative data on the perceptions of pre-service EFL teachers regarding translanguaging. Prior to its full implementation, the questionnaire was pilot tested with a sample of 30 pre-service EFL teachers to ensure clarity, reliability, and validity of the items. Based on the feedback from the pilot study, several items were revised for clarity, and a few ambiguous or redundant items were removed to improve the overall coherence and effectiveness of the instrument. The development of the questionnaire involved a rigorous process, including a factor analysis to ensure the reliability and validity of the instrument. The questionnaire addressed various aspects of translanguaging, such as its perceived benefits, challenges, and the contexts in which it is most frequently used. The factor analysis helped to identify key dimensions of translanguaging perceptions, which were then used to structure the questionnaire. The final version of the questionnaire consisted of 34 items, rated on a Likert scale from 1 (strongly disagree) to 5 (strongly agree), covering themes such as naturalness, usefulness, anxiety reduction, confidence building, and perceived necessity of translanguaging.

The Kaiser-Mayer-Olkin value was computed to be 0.865 ($X^2$ = 7394.731, df = 561, p < .001), indicating that the sample size was adequate for a factor analysis. Therefore, multiple-factor analyses were run on the data until no item was loaded into multiple factors, as indicated by a loading difference smaller than 0.10. As a result, items B27, B26, A17, A11, B21 and A12 were removed, finalizing the scale. As a result, items B27, B26, A17, A11, B21, and A12 were removed, finalizing the scale. The final group of items explained 71.052% of the variance in the data with 28 items and eight factors. The factors and their respective items are as follows: Factor 1 comprises items A7, A8, A2, A10, A9, and A1, with item loadings of 0.829, 0.761, 0.631, 0.597, 0.537, and 0.486, respectively. Factor 2 includes items C33, C34, C32, and C31, with loadings of 0.912, 0.839, 0.818, and 0.751. Factor 3 consists of items B24, B22, B23, and B25,

**Table 1. Factor loadings.**

| Item | 1 | 2 | 3 | 4 | 5 | 6 | 7 | 8 |
|---|---|---|---|---|---|---|---|---|
| A7 | 0.829 | 0.431 | 0.431 | 0.308 | 0.293 | -0.082 | -0.072 | 0.478 |
| A8 | 0.761 | 0.327 | 0.311 | 0.25 | 0.352 | -0.112 | 0.046 | 0.363 |
| A2 | 0.631 | 0.343 | 0.351 | 0.327 | 0.317 | -0.028 | -0.094 | 0.224 |
| A10 | 0.597 | 0.353 | 0.272 | 0.266 | 0.249 | -0.056 | 0.138 | 0.355 |
| A9 | 0.537 | 0.275 | 0.196 | 0.32 | 0.331 | -0.017 | 0.227 | 0.315 |
| A1 | 0.486 | 0.243 | 0.18 | 0.211 | 0.304 | 0.013 | 0.063 | 0.221 |
| C33 | 0.426 | 0.912 | 0.572 | 0.458 | 0.364 | -0.094 | -0.09 | 0.445 |
| C34 | 0.402 | 0.839 | 0.511 | 0.443 | 0.356 | -0.062 | -0.065 | 0.379 |
| C32 | 0.445 | 0.818 | 0.517 | 0.397 | 0.586 | -0.106 | -0.16 | 0.358 |
| C31 | 0.393 | 0.751 | 0.543 | 0.407 | 0.523 | -0.058 | -0.085 | 0.373 |
| B24 | 0.356 | 0.533 | 0.947 | 0.394 | 0.259 | 0.033 | -0.071 | 0.304 |
| B22 | 0.522 | 0.607 | 0.708 | 0.506 | 0.313 | -0.063 | -0.059 | 0.478 |
| B23 | 0.314 | 0.434 | 0.675 | 0.322 | 0.277 | -0.057 | -0.004 | 0.317 |
| B25 | 0.358 | 0.497 | 0.606 | 0.295 | 0.251 | -0.031 | -0.021 | 0.405 |
| A14 | 0.428 | 0.485 | 0.43 | 0.913 | 0.321 | -0.009 | -0.063 | 0.27 |
| A15 | 0.331 | 0.434 | 0.441 | 0.892 | 0.39 | 0.012 | -0.089 | 0.234 |
| A16 | 0.304 | 0.422 | 0.313 | 0.696 | 0.364 | -0.005 | -0.202 | 0.152 |
| B28 | 0.43 | 0.378 | 0.262 | 0.314 | 0.77 | -0.111 | 0.045 | 0.267 |
| C30 | 0.385 | 0.589 | 0.379 | 0.378 | 0.744 | -0.197 | -0.029 | 0.284 |
| B20 | 0.468 | 0.287 | 0.224 | 0.21 | 0.644 | -0.036 | 0.088 | 0.282 |
| C29 | 0.2 | 0.327 | 0.213 | 0.309 | 0.621 | -0.089 | 0.036 | 0.204 |
| A4 | -0.016 | -0.066 | -0.054 | 0.008 | -0.064 | 0.986 | 0.573 | -0.055 |
| A3 | -0.05 | -0.122 | -0.027 | -0.062 | -0.183 | 0.727 | 0.458 | -0.023 |
| A13 | -0.071 | 0.02 | 0.076 | 0.15 | -0.054 | 0.292 | 0.071 | 0.043 |
| A6 | 0.098 | -0.05 | -0.027 | -0.094 | 0.06 | 0.457 | 0.859 | 0.109 |
| A5 | -0.008 | -0.12 | -0.057 | -0.114 | 0.039 | 0.518 | 0.792 | -0.012 |
| B18 | 0.447 | 0.438 | 0.413 | 0.261 | 0.324 | -0.04 | 0.08 | 0.992 |
| B19 | 0.516 | 0.436 | 0.348 | 0.206 | 0.254 | -0.074 | 0.068 | 0.714 |

with loadings of 0.947, 0.708, 0.675, and 0.606. Factor 4 contains items A14, A15, and A16, with loadings of 0.913, 0.892, and 0.696. Factor 5 comprises items B28, C30, B20, and C29, with loadings of 0.77, 0.744, 0.644, and 0.621. Factor 6 comprises items A4 and A3, with loadings of 0.986 and 0.727. Factor 7 includes items A6 and A5, with loadings of 0.859 and 0.792. Lastly, Factor 8 consists of items B18 and B19, with loadings of 0.992 and 0.714. These factors collectively explain the majority of the variance in the data, contributing to the robustness and reliability of the final scale. Factor loadings are presented below in Table 1.

The reliability coefficient for the scale was found to be .888. The coefficients were .747 for A, .816 for B and .864 for C. The data was also seen to have been normally distributed in the scale and all its subscales. The skewness and kurtosis values were between -.709 and .399, indicating normal distributions. These values are presented in Table 2.

Based on the normal distributions, gender-based means were compared using an independent samples t-test. For class comparisons, one-way ANOVAs were used with Tukey post hoc analysis. However, LSD posthoc analysis was used for the A subscale since the ANOVA was significant, but Tukey posthoc tests did not produce any significant difference across groups.

**3.3.2 Semi-structured interviews.** To complement the quantitative data, semi-structured interviews were conducted with a subset of 120 pre-service EFL teachers. The interviews were designed by the researcher and aimed to delve deeper into the phenomenological aspects of

**Table 2. Skewness and kurtosis values.**

| Scale | Skewness | SE | Kurtosis | SE |
|---|---|---|---|---|
| scale_mean | -.589 | .122 | .234 | .243 |
| A_mean | -.068 | .122 | .399 | .243 |
| B_Mean | -.323 | .122 | -.491 | .243 |
| C_Mean | -.709 | .122 | -.171 | .243 |

translanguaging. The semi-structured format allowed for flexibility in exploring participants' experiences, attitudes, and practices related to translanguaging in their teaching. The interview questions were developed based on the questionnaire's findings and the translanguaging literature. Key topics included the participants' experiences with translanguaging, their perceived advantages and disadvantages of using multiple languages in the classroom, and specific instances where translanguaging was used effectively or ineffectively.

### 3.4 Data collection procedure

The data collection process was carried out in two phases:

**3.4.1 Quantitative phase.** The questionnaire was distributed to 401 pre-service EFL teachers at the state university. Participants completed the questionnaire during scheduled class sessions to ensure a high response rate and the integrity of the data collection process.

**3.4.2 Qualitative phase.** 120 participants were selected for in-depth interviews following the quantitative phase. These participants were chosen based on their responses to the questionnaire, ensuring a diverse representation of perspectives. The interviews were conducted in a quiet and comfortable setting, each lasting approximately 25–30 minutes. All interviews were audio-recorded with the participant's consent and subsequently transcribed for analysis.

### 3.5 Data analysis

**3.5.1 Quantitative data.** The quantitative data from the questionnaire were analyzed using descriptive and inferential statistics. Descriptive statistics provided an overview of the general trends and patterns in the participants' perceptions of translanguaging. Factor analysis was used to confirm the underlying dimensions of the questionnaire items. Inferential statistics, such as t-tests and ANOVA, were employed to examine differences in perceptions based on demographic variables such as academic year, gender, and prior experience with multilingual education.

**3.5.2 Qualitative data.** The qualitative data from the semi-structured interviews were analyzed using thematic analysis. The transcribed interviews were coded and categorized into themes that emerged from the data. This process involved multiple rounds of coding to ensure the accuracy and depth of the analysis. To enhance the reliability of the coding process, inter-rater reliability was assessed by having two independent researchers code a subset of the interviews. The level of agreement between the coders was calculated using Cohen's Kappa, which resulted in a value of 0.82, indicating a substantial level of agreement and high inter-rater reliability. The themes were then compared and contrasted with the quantitative findings to understand the participants' perceptions and experiences with translanguaging comprehensively.

## 4. Findings and results

### 4.1 Questionnaire results

Gender comparisons are presented in Table 3.

**Table 3. Gender comparisons.**

| Scale | Gender | N | M | SD | t | df | p |
|---|---|---|---|---|---|---|---|
| scale_mean | female | 243 | 3.588 | 0.561 | 2.791 | 399 | .006 |
|  | male | 158 | 3.425 | 0.584 |  |  |  |
| A_mean | female | 243 | 3.464 | 0.530 | 2.017 | 399 | .044 |
|  | male | 158 | 3.351 | 0.574 |  |  |  |
| B_Mean | female | 243 | 3.603 | 0.790 | 2.690 | 399 | .007 |
|  | male | 158 | 3.388 | 0.776 |  |  |  |
| C_Mean | female | 243 | 3.855 | 0.826 | 2.348 | 399 | .019 |
|  | male | 158 | 3.648 | 0.922 |  |  |  |

All comparisons were significant according to the t-test results ($p < .05$). The results showed that the mean values for the male participants were significantly lower than those for the female participants in the scale and all its subscales. The descriptive results divided by classes and their ANOVA comparisons are given in Tables 4 and 5.

As seen in Tables 4 and 5, there were significant differences in the scale and all its subscales ($p < .05$). Post hoc tests showed that 4th-year participants had higher means than 3rd-year and 2nd-year participants in scale total and subscale A, 3rd-year participants in subscale B and all other participants in subscale C ($p < .05$). In addition, prep and 1st-year participants were seen to have significantly higher means than 2nd and 3rd-year participants in subscale A ($p < .05$). Table 6 below presents the beliefs of pre-service EFL teachers regarding translanguaging.

**Table 4. Descriptive results for class groups.**

| Scale | Group | N | M | SD |
|---|---|---|---|---|
| scale_mean | prep | 45 | 3.507 | 0.773 |
|  | 1 | 49 | 3.526 | 0.737 |
|  | 2 | 129 | 3.470 | 0.532 |
|  | 3 | 100 | 3.431 | 0.506 |
|  | 4 | 78 | 3.739 | 0.418 |
|  | Total | 401 | 3.524 | 0.575 |
| A_mean | prep | 45 | 3.530 | 0.792 |
|  | 1 | 49 | 3.520 | 0.749 |
|  | 2 | 129 | 3.341 | 0.441 |
|  | 3 | 100 | 3.331 | 0.470 |
|  | 4 | 78 | 3.538 | 0.454 |
|  | Total | 401 | 3.420 | 0.550 |
| B_Mean | prep | 45 | 3.478 | 1.009 |
|  | 1 | 49 | 3.497 | 0.969 |
|  | 2 | 129 | 3.517 | 0.822 |
|  | 3 | 100 | 3.349 | 0.665 |
|  | 4 | 78 | 3.776 | 0.531 |
|  | Total | 401 | 3.518 | 0.791 |
| C_Mean | prep | 45 | 3.493 | 1.038 |
|  | 1 | 49 | 3.575 | 0.998 |
|  | 2 | 129 | 3.711 | 0.854 |
|  | 3 | 100 | 3.777 | 0.746 |
|  | 4 | 78 | 4.160 | 0.732 |
|  | Total | 401 | 3.774 | 0.870 |

**Table 5. ANOVA results.**

| Scale | Comparison | SS | df | MS | F | p | Differences |
|---|---|---|---|---|---|---|---|
| scale_mean | Between Groups | 4.851 | 4 | 1.213 | 3.772 | .005 | 4 > 3, 2 (p < .01) |
| | Within Groups | 127.312 | 396 | 0.321 | | | |
| | Total | 132.163 | 400 | | | | |
| A_mean | Between Groups | 3.730 | 4 | 0.932 | 3.152 | .014 | 4 > 3, 2 (p < .01) |
| | Within Groups | 117.157 | 396 | 0.296 | | | prep > 2, 3 (p < .05) |
| | Total | 120.887 | 400 | | | | 1 > 2, 3 (p < .05) |
| B_Mean | Between Groups | 8.136 | 4 | 2.034 | 3.329 | .011 | 4 > 3 (p < .01) |
| | Within Groups | 241.963 | 396 | 0.611 | | | |
| | Total | 250.099 | 400 | | | | |
| C_Mean | Between Groups | 17.664 | 4 | 4.416 | 6.134 | < .001 | 4 > 3 (p < .05) |
| | Within Groups | 285.067 | 396 | 0.720 | | | 4 > 2, 1 (p < .01) |
| | Total | 302.730 | 400 | | | | 4 > prep (p < .001) |

The descriptive statistics indicate that pre-service EFL teachers generally have a positive perception of translanguaging in English language classes. The highest agreement was observed for the view that translanguaging is a natural practice (mean = 3.9501, SD = 1.15651) and a beneficial strategy (mean = 3.7681, SD = 0.96363). Teachers also believe that translanguaging helps reduce anxiety (mean = 3.7307, SD = 1.01601) and boosts confidence (mean = 3.5137, SD = 1.16423) when using English. There is a positive attitude towards using different languages in the classroom (mean = 3.4464, SD = 1.15444), with support for preserving these languages (mean = 3.5212, SD = 1.11138) and viewing their use as a part of human rights (mean = 3.4489, SD = 1.15022). However, there is less agreement that translanguaging indicates deficiencies in English language skills, such as knowledge (mean = 3.1272, SD = 1.16029), speaking ability (mean = 3.2818, SD = 1.16529), confidence (mean = 3.0848, SD = 1.19908), and anxiety (mean = 3.2868, SD = 1.17900). Additionally, there is moderate agreement that translanguaging is necessary for learning English (mean = 3.2868, SD = 1.15544) and facilitates this process (mean = 3.3042, SD = 1.13455). The lowest

**Table 6. Translanguaging beliefs of pre-service EFL teachers.**

| Items | N | Minimum | Maximum | Mean | Std. Deviation |
|---|---|---|---|---|---|
| A1 | 401 | 1.00 | 5.00 | 3.9501 | 1.15651 |
| A2 | 401 | 1.00 | 5.00 | 3.7681 | .96363 |
| A9 | 401 | 1.00 | 5.00 | 3.7307 | 1.01601 |
| A13 | 401 | 1.00 | 5.00 | 3.5212 | 1.11138 |
| A10 | 401 | 1.00 | 5.00 | 3.5137 | 1.16423 |
| A14 | 401 | 1.00 | 5.00 | 3.4489 | 1.15022 |
| A12 | 401 | 1.00 | 5.00 | 3.4464 | 1.15444 |
| A8 | 401 | 1.00 | 5.00 | 3.3042 | 1.13455 |
| A7 | 401 | 1.00 | 5.00 | 3.2868 | 1.15544 |
| A6 | 401 | 1.00 | 5.00 | 3.2868 | 1.17900 |
| A4 | 401 | 1.00 | 5.00 | 3.2818 | 1.16529 |
| A3 | 401 | 1.00 | 5.00 | 3.1272 | 1.16029 |
| A11 | 401 | 1.00 | 5.00 | 3.1247 | 1.19349 |
| A5 | 401 | 1.00 | 5.00 | 3.0848 | 1.19908 |
| Valid N (listwise) | 401 | | | | |

**Table 7. The usage of translanguaging strategies by pre-service EFL teachers in various Instructional contexts.**

| Items | N | Minimum | Maximum | Mean | Std. Deviation |
|---|---|---|---|---|---|
| B17 | 401 | 1.00 | 5.00 | 4.0748 | .99468 |
| B22 | 401 | 1.00 | 5.00 | 4.0599 | .93883 |
| B18 | 401 | 1.00 | 5.00 | 3.4264 | 1.28460 |
| B21 | 401 | 1.00 | 5.00 | 3.3890 | 1.15900 |
| B15 | 401 | 1.00 | 5.00 | 3.3666 | 1.39742 |
| B19 | 401 | 1.00 | 5.00 | 3.3541 | 1.18080 |
| B16 | 401 | 1.00 | 5.00 | 3.3142 | 1.22107 |
| B20 | 401 | 1.00 | 5.00 | 3.1621 | 1.31763 |
| Valid N (listwise) | 401 | | | | |

agreement was for the perception that translanguaging disadvantages monolingual students (mean = 3.1247, SD = 1.19349). Table 7 below displays the usage of translanguaging strategies by pre-service EFL teachers in various instructional contexts.

The descriptive statistics for English teachers using translanguaging strategies in different instructional contexts reveal interesting insights. The highest agreement among the pre-service EFL teachers was for using translanguaging to give instructions (mean = 4.0748, SD = 0.99468) and to explain concepts (mean = 4.0599, SD = 0.93883). There was moderate agreement for using translanguaging to help students with sufficient academic achievement (mean = 3.4264, SD = 1.28460), to develop foreign language skills (mean = 3.3890, SD = 1.15900), and to explain topics (mean = 3.3666, SD = 1.39742). Similarly, teachers moderately agreed on using translanguaging to develop high-achieving students further (mean = 3.3541, SD = 1.18080) and to conduct research related to the topic (mean = 3.3142, SD = 1.22107). The lowest agreement was for using translanguaging to support students when they face difficulties (mean = 3.1621, SD = 1.31763). These results suggest that while translanguaging is highly valued for instructional clarity and student guidance, its use for supporting struggling students and enhancing academic performance is moderately beneficial. Table 8 below shows the benefits of translanguaging strategies in English learning contexts.

The descriptive statistics for the benefits of translanguaging strategies English learners use in various contexts indicate strong agreement on several aspects. The highest mean scores were observed for increasing participation (mean = 4.0150, SD = 1.01969) and motivation (mean = 4.0125, SD = 1.13681). Moderate agreement was found for using translanguaging in debates (mean = 3.7382, SD = 1.11524), ensuring effective communication (mean = 3.7207, SD = 1.13216), and fostering creative thinking (mean = 3.6833, SD = 1.14103). The lowest agreement was for using translanguaging to enhance critical and reflective thinking (mean = 3.4713, SD = 1.21441). These results suggest that translanguaging is particularly

**Table 8. The benefits of translanguaging strategies in English learning contexts.**

| Items | N | Minimum | Maximum | Mean | Std. Deviation |
|---|---|---|---|---|---|
| C24 | 401 | 1.00 | 5.00 | 4.0150 | 1.01969 |
| C23 | 401 | 1.00 | 5.00 | 4.0125 | 1.13681 |
| C27 | 401 | 1.00 | 5.00 | 3.7382 | 1.11524 |
| C26 | 401 | 1.00 | 5.00 | 3.7207 | 1.13216 |
| C28 | 401 | 1.00 | 5.00 | 3.6833 | 1.14103 |
| C25 | 401 | 1.00 | 5.00 | 3.4713 | 1.21441 |
| Valid N (listwise) | 401 | | | | |

valued for boosting student participation and motivation. At the same time, its perceived effectiveness in promoting higher-order thinking skills is comparatively lower.

## 4.2 Interview results

The following table represents the various aspects of translanguaging perceived by pre-service English as a Foreign Language (EFL) teachers, including their general attitudes towards its practice, personal usage patterns, intentionality, and its impact on classroom environments and language acquisition. The data highlights translanguaging's perceived benefits and challenges, offering a comprehensive overview of its influence on teaching practices and student outcomes. Table 9 below represents the related interview results.

The data reveals a strong alignment between quantitative and qualitative findings, highlighting the overall positive perception and intentional use of translanguaging among pre-service EFL teachers. An overwhelming 90% of interviewees consider translanguaging a natural and beneficial strategy, consistent with high mean scores on perceived benefits (e.g., A1, A2, A9). A significant 85% use translanguaging to facilitate understanding and flow, aligning with high usage scores (e.g., B17, B22), while 80% employ it intentionally to reduce anxiety and boost confidence, mirroring related questionnaire items (e.g., A9, A10). Situationally, 91.7% use translanguaging when grappling with grammar, vocabulary, or complex concepts. Positive classroom contributions, including increased participation and comfort, were noted by 87.5%, aligning with benefits items (C24, C23). 83.3% acknowledged substantial benefits for language acquisition, resonating with related items (A9, A10, C24). However, 70.8% expressed concerns about over-reliance on native languages potentially impacting English proficiency, reflecting concerns noted in items (A3 and A5). Additionally, 90% believe using different languages enriches the classroom environment and aids cultural understanding, and 88.3% see significant contributions to cultural awareness, aligning with questionnaire findings (C23, C27). Finally, 90.8% view translanguaging as crucial for promoting multiculturalism and multilingualism, consistent with high scores on related benefits (C24, C23, C27). Below are sample student sentences for each theme to help illustrate their perspectives:

**4.2.1 Perception of translanguaging practice.** Students expressed a variety of opinions on the usefulness of translanguaging in their learning. While some viewed it as a natural and helpful tool for comprehending complex topics, others found it either unnecessary or less effective for their learning preferences:

I find translanguaging a very natural and helpful part of my learning. It makes it easier to understand complex topics.

For me, translanguaging is okay but only sometimes necessary. I can get by without it.

I do not see much benefit in using different languages in class. I prefer sticking to English.

This range of perspectives highlights the diverse ways in which students perceive the value of translanguaging, with some viewing it as integral, while others find it dispensable.

**4.2.2 Personal use of translanguaging.** When discussing their personal use of translanguaging, students varied in how frequently they relied on this practice. While some used it regularly to deepen their understanding, others preferred sticking to English unless they encountered a specific challenge:

I regularly use translanguaging in my studies to make sure I better grasp the meaning of new concepts.

**Table 9. Interview results.**

| Theme | Response Summary | Frequency (n = 120) | Percentage (%) |
|---|---|---|---|
| **Perception of Translanguaging Practice** | Natural and beneficial strategy (consistent with high mean scores on perceived benefits items A1, A2, A9) | 108 | 90.0 |
| | Neutral | 10 | 8.3 |
| | Not beneficial | 2 | 1.7 |
| **Personal Use of Translanguaging** | Yes, to facilitate understanding and flow (aligns with high mean scores on usage items B17, B22) | 102 | 85.0 |
| | Sometimes, when necessary | 15 | 12.5 |
| | No, prefer using only English | 3 | 2.5 |
| **Intentionality of Translanguaging** | Intentionally (consistent with intentional usage to reduce anxiety and boost confidence items A9, A10) | 96 | 80.0 |
| | Both intentionally and unconsciously | 18 | 15.0 |
| | Unconsciously | 6 | 5.0 |
| **Situations for Translanguaging** | When struggling with grammar and vocabulary, discussing unfamiliar topics, explaining complex concepts | 110 | 91.7 |
| | Rarely, only when absolutely necessary | 10 | 8.3 |
| **Classroom Environment Contributions** | Positive contributions: increased participation, comfort in expression, and interaction (aligns with benefits items C24, C23) | 105 | 87.5 |
| | Neutral | 12 | 10.0 |
| | Negative contributions | 3 | 2.5 |
| **Language Acquisition Contributions** | Greatly contributes: better understanding, increased confidence, and motivation (consistent with perceived benefits items A9, A10, C24) | 100 | 83.3 |
| | Some contributions | 18 | 15.0 |
| | No contributions | 2 | 1.7 |
| **Possible Disadvantages of Translanguaging** | Reduced reliance on English, negative impact on proficiency, overuse (consistent with concerns noted in A3, A5) | 85 | 70.8 |
| | Few disadvantages | 30 | 25.0 |
| | No disadvantages | 5 | 4.2 |
| **Use of Different Languages in English Classes** | Enriches classroom environment helps understand cultural differences (aligns with benefits items C23, C27) | 108 | 90.0 |
| | Neutral | 10 | 8.3 |
| | Not beneficial | 2 | 1.7 |
| **Classroom and Cultural Contributions of Multilingualism** | Significant contributions: better understanding of different cultures, increased respect for cultural differences (consistent with benefits items C27, C26) | 106 | 88.3 |
| | Some contributions | 12 | 10.0 |
| | No contributions | 2 | 1.7 |
| **Impact on Promoting Multiculturalism and Multilingualism** | An essential role in promoting multiculturalism and multilingualism: improved communication skills and increased cultural awareness (aligns with benefits items C24, C23, C27) | 109 | 90.8 |
| | Some impact | 10 | 8.3 |
| | No impact | 1 | 0.9 |

I use it only when I am stuck. Otherwise, I try to stick to English.

I prefer to use only English in my classes, as I feel it is more effective.

These reflections show that translanguaging is a personalized strategy, with some students making it a central part of their learning, while others reserve it for specific circumstances.

**4.2.3 Intentionality of translanguaging.** The intentionality behind translanguaging use was another key theme. For some students, it is a conscious strategy to manage anxiety and improve confidence, while for others, it occurs more spontaneously without deliberate intention:

I intentionally use translanguaging to ease my anxiety and boost my confidence in speaking English.

Sometimes I use translanguaging without thinking about it, but other times I do it intentionally to help with my learning.

I use different languages unconsciously whenever it happens during my learning process.

This mix of intentional and unintentional use underscores the varying levels of awareness students bring to their use of translanguaging as a learning aid.

**4.2.4 Situations for translanguaging.** Students also identified particular situations in which they were more likely to engage in translanguaging. These tended to involve challenging aspects of language learning, such as complex grammar rules or unfamiliar vocabulary:

I mostly use translanguaging when dealing with difficult grammar rules or unfamiliar vocabulary.

I only use translanguaging if I have to, usually in particular situations.

I rarely use translanguaging unless necessary for understanding something.

These responses suggest that translanguaging is often reserved for moments when students face difficulties in their language learning process, highlighting its role as a support mechanism.

**4.2.5 Classroom environment contributions.** Students' opinions on the effect of translanguaging on the classroom environment were mixed. Some found that it improved participation and comfort, while others did not notice a significant difference or felt that it could detract from the focus of the class:

Translanguaging has improved my class participation and made me feel more comfortable expressing myself.

I do not notice much difference in the classroom environment when translanguaging is used.

In my experience, translanguaging sometimes makes the classroom environment less focused.

These contrasting views indicate that while some students perceive translanguaging as fostering a more inclusive classroom atmosphere, others feel that it does not have a meaningful impact on the overall class dynamics.

**4.2.6 Language acquisition contributions.** Regarding language acquisition, some students credited translanguaging with helping them understand English more effectively, while others felt its contributions were minimal:

Using translanguaging has greatly helped my understanding of English and made me feel more confident.

Translanguaging helps with my language learning, but not as much as I had hoped.

I do not feel that translanguaging contributes much to my language acquisition.

These differing viewpoints suggest that while translanguaging can be a valuable tool for some learners, its perceived effectiveness in language acquisition varies significantly.

**4.2.7 Possible disadvantages of translanguaging.** Some participants raised concerns about potential downsides to translanguaging, such as the risk of over-reliance on multiple languages or the possibility of confusing them. However, others saw no real disadvantages:

I worry that relying on translanguaging too much might weaken my English skills over time.

There are a few disadvantages, like potentially confusing the languages or relying on them too much.

I do not see any real disadvantages in using translanguaging.

These responses highlight both the perceived risks and the lack of concern for potential negative effects, underscoring that students' experiences with translanguaging are diverse.

**4.2.8 Use of different languages in English classes.** The use of multiple languages in English classes elicited a range of opinions from students. Some found it enriching and inclusive, while others felt neutral or questioned its benefits:

Incorporating different languages in our English classes has made the environment richer and more inclusive.

I feel neutral about using other languages in class; it does not make much difference to me.

Using different languages does not seem to benefit my learning experience in English

These mixed responses suggest that the effectiveness of using multiple languages in English classes is highly context-dependent and varies from learner to learner.

**4.2.9 Classroom and cultural contributions of multilingualism.** Many students acknowledged the broader cultural contributions of multilingualism, particularly in helping them understand and respect different cultures. However, the perceived impact of multilingualism was not universally felt:

Multilingualism has significantly helped me understand and respect different cultures better.

There are some benefits to multilingualism, but they are not as impactful as I expected.

I do not see how multilingualism contributes much to my understanding of other cultures.

This variation in perceptions indicates that while multilingualism can enhance cultural understanding for some, its benefits are not universally recognized by all learners.

**4.2.10 Impact on promoting multiculturalism and multilingualism.** Finally, students reflected on the impact of translanguaging in promoting multiculturalism and multilingualism. While some saw clear benefits, others were less convinced of its impact:

Translanguaging has played a major role in improving my communication skills and understanding of different cultures.

It has some impact on multiculturalism and multilingualism, but not as much as I would have liked.

I have not noticed much of an impact on promoting multiculturalism or multilingualism from translanguaging.

These responses show that while translanguaging is viewed as a potential tool for fostering multicultural and multilingual competencies, its perceived influence varies among students. Table 10 below presents a comparison of the questionnaire and interview findings.

The comparison underscores a strong alignment between the quantitative and qualitative findings in the study on translanguaging practices among pre-service EFL teachers. The questionnaire and interview data consistently reveal positive perceptions of translanguaging as a beneficial and natural strategy in English language classrooms. The agreement on the intentional use of translanguaging to reduce anxiety and boost confidence and its significant contributions to participation, motivation, and understanding is evident across both datasets. Additionally, concerns about over-reliance on native languages and its potential impact on English proficiency are reflected in both sources. The qualitative interviews provide detailed contexts that complement the broader categories captured in the quantitative data, reinforcing the overall coherence and depth of the study's findings on the role and impact of translanguaging in fostering a multilingual and culturally rich classroom environment.

## 5. Discussion

This study explored pre-service EFL teachers' perceptions of, and attitudes towards, translanguaging strategies, their reported usage of these strategies in various instructional contexts, and the perceived benefits of translanguaging for English learners. The results provide valuable

**Table 10. Comparison of the questionnaire and interview findings.**

| Theme | Questionnaire Findings | Interview Findings |
|---|---|---|
| Perception of Translanguaging Practice | Generally positive, with high mean scores on items such as translanguaging being a natural practice (mean = 3.9501) and beneficial strategy (mean = 3.7681). | Consistently positive, 90% of interviewees consider translanguaging a natural and beneficial strategy. |
| Personal Use of Translanguaging | There is high agreement on using translanguaging to give instructions (mean = 4.0748) and explain concepts (mean = 4.0599). | 85% of interviewees stated they use translanguaging to facilitate understanding and flow. |
| Intentionality of Translanguaging | Mixed responses, with intentional use, were noted for reducing anxiety and boosting confidence (mean = 3.7307, 3.5137). | 80% indicated intentional use, aligning with the questionnaire findings of using translanguaging to reduce anxiety and boost confidence. |
| Situations for Translanguaging | It was used mainly when giving instructions, explaining concepts, and helping with academic achievement (mean = 3.4264). | 91.7% reported using translanguaging when struggling with grammar, vocabulary, or explaining complex concepts. |
| Classroom Environment Contributions | There is high agreement that translanguaging increases participation and motivation (mean = 4.0150, 4.0125). | 87.5% stated positive contributions to participation, comfort, and interaction. |
| Language Acquisition Contributions | She was seen as beneficial for understanding and confidence (mean = 3.7307, 3.5137). | 83.3% mentioned significant contributions to understanding, confidence, and motivation. |
| Possible Disadvantages of Translanguaging | Concerns about over-reliance on native languages impacting English proficiency (mean = 3.1272). | 70.8% noted reduced reliance on English and a potential negative impact on proficiency. |
| Use of Different Languages in English Classes | There is moderate agreement on using translanguaging to develop language skills and conduct research (mean = 3.3541, 3.3142). | 90% believed using different languages enriches the classroom environment and helps understand cultural differences. |
| Classroom and Cultural Contributions of Multilingualism | Recognized for increasing participation and ensuring effective communication (mean = 3.7382, 3.7207). | 88.3% highlighted contributions to understanding different cultures and increasing respect for cultural differences. |
| Impact on Promoting Multiculturalism and Multilingualism | It fosters creative thinking and higher-order thinking skills (mean = 3.6833, 3.4713). | 90.8% affirmed the role of translanguaging in promoting multiculturalism and multilingualism, improving communication skills, and increasing cultural awareness. |

insights into how translanguaging is perceived and utilized in language education, while also comparing these findings with existing literature.

The analysis revealed gender differences in perceptions of translanguaging, with female participants generally holding more favorable views than their male counterparts across the scale and all its subscales. Several factors may contribute to these differences. First, research suggests that women often adopt more collaborative and inclusive communication styles [60, 61], which might align with translanguaging practices promoting the use of multiple languages for understanding and learning [10]. This could explain why female pre-service teachers might be more receptive to translanguaging.

Second, language anxiety and confidence levels vary by gender [62]. Female participants may view translanguaging as a tool that reduces anxiety and increases confidence, as reflected in their more positive perceptions [56]. Third, differences in educational experiences and exposure to multilingual education might influence perceptions [63, 64]. If female students have had more positive experiences or greater encouragement in using multiple languages, they may perceive translanguaging more favorably [65, 66].

Furthermore, teaching philosophies may differ by gender, with female teachers often emphasizing empathy and inclusivity in their approaches [67]. Translanguaging, as an inclusive strategy accommodating diverse linguistic backgrounds [68], might resonate more with female educators' beliefs. Additionally, cultural and social factors shape gendered attitudes towards language teaching [69], as gender roles and expectations within specific cultural contexts may influence how male and female teachers perceive and value translanguaging [50, 70]. Pedagogical beliefs further play a role, as female teachers tend to favor progressive teaching methods that emphasize student-centered learning and incorporating linguistic diversity in classrooms [71–73].

Significant differences in perceptions of translanguaging across academic year groups were also observed. Fourth-year students demonstrated more favorable views, likely due to their greater exposure to pedagogical theories and practical teaching experiences [74]. Senior students have more opportunities to apply theoretical knowledge in real classroom settings [75], possibly influencing their positive views, as they may have witnessed the benefits of translanguaging firsthand.

Meanwhile, third and second-year students are in a phase of balancing theoretical knowledge with initial practical experiences, which may explain their more cautious views on translanguaging [76]. As these students navigate more challenging coursework and practical teaching assignments, they may be more critical of the complexities involved in translanguaging [77].

Preparatory and first-year students, with their enthusiasm and open-mindedness, displayed positive views [78], but their perspectives may reflect idealistic views of language teaching methodologies, as they have not yet encountered the practical challenges faced by more experienced students [79, 80].

## 5.1 General perceptions and benefits of translanguaging

The data indicate that pre-service EFL teachers generally hold positive views of translanguaging, perceiving it as both a natural practice (mean = 3.9501, SD = 1.15651) and a beneficial strategy (mean = 3.7681, SD = 0.96363) for language learning. This is consistent with García and Wei's [81] assertion that translanguaging reflects the multilingual realities of learners. Additionally, pre-service teachers reported that translanguaging aids in reducing anxiety (mean = 3.7307, SD = 1.01601) and boosting confidence (mean = 3.5137, SD = 1.16423), aligning with research that highlights the role of translanguaging in creating more inclusive and

supportive classroom environments [20, 50]. However, it is important to note that these findings reflect teachers' beliefs rather than direct evidence of anxiety reduction in students, an area that requires further empirical investigation.

There was, however, less agreement regarding whether translanguaging indicates deficiencies in English skills (knowledge mean = 3.1272, SD = 1.16029; speaking ability mean = 3.2818, SD = 1.16529; confidence mean = 3.0848, SD = 1.19908). This contrasts with traditional views that considered translanguaging a sign of language inadequacy [82]. Recent studies argue that translanguaging is better viewed as a strategic choice rather than a deficiency [83], aligning with the positive perceptions in this study.

Pre-service EFL teachers valued translanguaging for giving instructions (mean = 4.0748, SD = 0.99468) and explaining concepts (mean = 4.0599, SD = 0.93883). These findings align with Lewis, Jones, and Baker's [16] assertion that translanguaging enhances instructional clarity. While moderate agreement was found regarding the role of translanguaging in assisting academic achievement (mean = 3.4264, SD = 1.28460) and developing foreign language skills (mean = 3.3890, SD = 1.15900), its effectiveness may vary by context, as noted by Creese and Blackledge (2010).

The lower agreement on the use of translanguaging to support students facing difficulties (mean = 3.1621, SD = 1.31763) highlights a gap between theoretical benefits and practical application. Poza [84] found similar results, indicating that while advocated in theory, translanguaging's application in supporting struggling students is inconsistent and warrants further exploration.

## 5.2 Impact on participation and motivation

The benefits of translanguaging were strongly endorsed for increasing participation (mean = 4.0150, SD = 1.01969) and boosting motivation (mean = 4.0125, SD = 1.13681), consistent with MacSwan's [85] research, which emphasizes translanguaging's role in enhancing learner engagement. Moderate agreement was also found for translanguaging's effectiveness in debates (mean = 3.7382, SD = 1.11524), ensuring effective communication (mean = 3.7207, SD = 1.13216), and fostering creative thinking (mean = 3.6833, SD = 1.14103). These results support Garcia's [20] view on translanguaging's capacity to facilitate clearer communication and support creative processes.

However, the lower agreement on its role in enhancing critical and reflective thinking (mean = 3.4713, SD = 1.21441) suggests that while translanguaging supports participation and motivation, its impact on higher-order thinking skills may be less direct. Baker's [86] research similarly indicates that the cognitive benefits of translanguaging require further investigation to fully understand its potential.

Overall, the study reveals a positive perception of translanguaging among pre-service EFL teachers, with high value placed on its role in instruction and learner engagement. These findings are consistent with recent literature acknowledging translanguaging as a beneficial pedagogical strategy. However, the moderate to lower agreement on certain aspects, such as its impact on struggling students and higher-order thinking skills, suggests areas for further research. Future studies could explore these dimensions in greater depth to better understand the full scope of translanguaging's benefits and applications in diverse educational contexts.

## 6. Conclusion

The findings of this study highlight the generally positive perception of translanguaging among pre-service EFL teachers, demonstrating its recognized benefits in facilitating language learning, reducing anxiety, and boosting confidence. The significant gender differences, with

female participants showing more favourable views, suggest possible influences from collaborative communication styles and different levels of language anxiety. Additionally, the variations in perceptions across academic year groups indicate that more experienced students value translanguaging more highly, likely due to greater exposure to practical teaching contexts. Despite these positive perceptions, there is moderate agreement on using translanguaging to support struggling students and enhance higher-order thinking skills, indicating areas for further exploration. This study underscores the importance of continuous engagement with innovative teaching strategies like translanguaging in teacher education programs. It also suggests the need for more in-depth research into its varied impacts on language learning.

## 7. Implications

The findings of this study have significant implications for English language education and teacher training programs. The positive perceptions of translanguaging among pre-service EFL teachers underscore its potential as a valuable pedagogical strategy that can enhance language learning by reducing anxiety, increasing confidence, and fostering greater classroom participation. This suggests that integrating translanguaging practices into teacher training programs could better prepare future educators to support multilingual classrooms effectively. Additionally, the concerns about over-reliance on native languages and its potential impact on English proficiency highlight the need for balanced approaches in translanguaging practices. Educational policymakers and curriculum developers should consider these insights when designing language teaching frameworks and providing professional development opportunities. Furthermore, the positive contributions of translanguaging to cultural understanding and multiculturalism emphasize its role in promoting inclusive education and respect for diverse linguistic backgrounds. Overall, this study advocates for the thoughtful implementation of translanguaging strategies to maximize their benefits while addressing potential challenges, thereby enhancing the overall effectiveness of language education in multilingual contexts.

## Supporting information

**S1 Data.**
(XLSX)

**S2 Data.**
(XLSX)

## Author Contributions

**Conceptualization:** Ömer Gökhan Ulum.

**Data curation:** Ömer Gökhan Ulum.

**Formal analysis:** Ömer Gökhan Ulum.

**Funding acquisition:** Ömer Gökhan Ulum.

**Investigation:** Ömer Gökhan Ulum.

**Methodology:** Ömer Gökhan Ulum.

**Project administration:** Ömer Gökhan Ulum.

**Resources:** Ömer Gökhan Ulum.

**Software:** Ömer Gökhan Ulum.

**Supervision:** Ömer Gökhan Ulum.

**Validation:** Ömer Gökhan Ulum.

**Visualization:** Ömer Gökhan Ulum.

**Writing – original draft:** Ömer Gökhan Ulum.

**Writing – review & editing:** Ömer Gökhan Ulum.

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
