## [Decision Letter · Decision Letter 0]

4 Oct 2024

PONE-D-24-31275Breaking Down Linguistic Barriers: The Radical Impact of Translanguaging on Pre-Service EFL Teachers' Perspectives in TurkeyPLOS ONE

Dear Dr. Ulum,

Thank you for submitting your manuscript to PLOS ONE. After careful consideration, we feel that it has merit but does not fully meet PLOS ONE’s publication criteria as it currently stands. Therefore, we invite you to submit a revised version of the manuscript that addresses the points raised during the review process.

The reviewers have many nice things to say about the manuscript but feel that a few areas need more clarity. Then I have made some editorial suggestions regarding the abstract and the citations.  So please look at the journal formatting and fix the citations. 

We look forward to receiving your revised manuscript.

Kind regards,

Mary Diane Clark, PhD

Academic Editor

PLOS ONE

Additional Editor Comments:

Thank you for your article. the reviewers have a few changes that they want to see to improve the manuscript. I have attached your manuscript with some strike outs in the abstract --no data here.

Then please look up how citations happen in PLOS ONE journals. The use numbers in text so please follow their formatting.

Reviewers' comments:

Reviewer's Responses to Questions

**Comments to the Author**

1. Is the manuscript technically sound, and do the data support the conclusions?

Reviewer #1: Yes

Reviewer #2: Yes

2. Has the statistical analysis been performed appropriately and rigorously? 

Reviewer #1: I Don't Know

Reviewer #2: Yes

3. Have the authors made all data underlying the findings in their manuscript fully available?

Reviewer #1: Yes

Reviewer #2: Yes

4. Is the manuscript presented in an intelligible fashion and written in standard English?

Reviewer #1: Yes

Reviewer #2: Yes

5. Review Comments to the Author

Reviewer #1: What do you mean by valuable and natural pedagogical strategy? Please clarify.

p. 3: The history of translanguaging needs a fuller description.

By way of history, Zhang (2022) notes that Colin Baker coined the term translanguaging by placing the trans prefix to the term languaging when he was introduced to the work of Cen Williams (2000), leader of Welsh language revitalization programs. Subsequently, the notion of translanguaging has developed as a sociolinguistic theory (Otheguy et al., 2015) and an innovative teaching practice (Canagarajah, 2022).

p. 7: Citations needed for language anxiety is a significant barrier to language acquisition.

p. 11-12: was the questionnaire pilot tested?

Otheguy, R., García, O., & Reid, W. (2015). Clarifying translanguaging and deconstructing

named languages: A perspective from linguistics. Applied Linguistics Review, 6(3), 281-307.

Zhang, L. J. (2022). Deepening the understanding of translanguaging as a practical theory of language: A conversation with Professor Li Wei. RELC Journal, 53(3), 739-7446.)

Reviewer #2: This is a clear and well written study of the use of translanguaging in the Turkish context. I have only a few suggestions for the authors:

1. I would love a few more details about the qualitative coding, specifically whether any inter rater reliability was attempted or attained

2. This may be a personal preference, but I do not like the presentation of a list of isolated quotes for qualitative results - (pp. 23-26). I'd love some author input between the quotes to provide context/situational description.

3. I think the first paragraph of the discussion (pages. 29-31) could benefit from paragraphs to organize the various hypotheses presented - having them all in one paragraph was a bit overwhelming. Because there are several hypotheses presented for the gender differences, I think it would be more readable to organize them in this way together

4. I think more caution needs to be taken on page 31-32 connecting to Auerbach's findings because this study does not examine whether lowered anxiety actually happens, but rather teacher's beliefs that it could happen

5. The references list did not appear to follow APA formatting for me with regard to the use of italics, etc. May have been a formatting issue in the submission but probably needs to be addresse.d

6. PLOS authors have the option to publish the peer review history of their article (what does this mean?). If published, this will include your full peer review and any attached files.

Reviewer #1: No

Reviewer #2: No

---

## [Author Response · Author response to Decision Letter 0]

9 Oct 2024

Dear Editor and Reviewers, 

Thank you for your insightful and constructive feedback throughout the review process. Your detailed comments and suggestions have greatly enhanced the clarity and rigor of this manuscript. I have carefully addressed all points raised, and I hope the revisions now meet the journal's standards. Your guidance has been invaluable in improving the quality of this work. 

Editor: Then I have made some editorial suggestions regarding the abstract and the citations. So please look at the journal formatting and fix the citations.

I have revised the abstract and citations and I have also reviewed the journal formatting guidelines and have made the necessary adjustments to ensure that all citations are correctly formatted according to the journal's requirements.

Reviewer #1: 

Reviewer #1: What do you mean by valuable and natural pedagogical strategy? Please clarify.

1. 

The reviewer asked what I mean by valuable and natural pedagogical strategy in this part of the abstract ‘’The results indicate that translanguaging is largely perceived as a valuable and natural pedagogical strategy that alleviates anxiety, enhances confidence, and promotes active classroom participation.’’

I clarified this ambiguity as in the bold part and added it into original text. 

The results suggest that translanguaging is widely perceived as a valuable and natural pedagogical strategy, as it enhances learning by reducing language anxiety, fostering inclusivity, and promoting active participation. Its "natural" aspect lies in the alignment with learners' spontaneous use of their full linguistic repertoire, facilitating smoother integration into the learning process. Translanguaging is shown to alleviate anxiety, build confidence, and stimulate classroom engagement.

2. 

The reviewer stated the history of translanguaging needs a fuller description. and added the following paragraph 

By way of history, Zhang (2022) notes that Colin Baker coined the term translanguaging by placing the trans prefix to the term languaging when he was introduced to the work of Cen Williams (2000), leader of Welsh language revitalization programs. Subsequently, the notion of translanguaging has developed as a sociolinguistic theory (Otheguy et al., 2015) and an innovative teaching practice (Canagarajah, 2022).

I added the following historical description taking the paragraph into consideration and enriching it accordingly. 

The concept of translanguaging, initially developed within the context of bilingual education, has its origins in the work of Welsh scholars during the 1980s. Colin Baker is credited with coining the term translanguaging when he encountered the work of Cen Williams, a leader in Welsh language revitalization programs (Lewis et al., 2012a; Lewis et al., 2012b; Zhang, 2022). Originally applied to Welsh-English bilingual classrooms, translanguaging referred to the systematic use of both languages for different instructional purposes (Jones, 2017). The practice allowed students to engage in a dynamic interchange between languages, with the aim of promoting deeper understanding and greater cognitive engagement through both languages. As translanguaging evolved, it began to challenge the traditional notion of language separation, which had long dominated bilingual education (García & Lin, 2017). Instead of viewing languages as distinct and isolated entities, translanguaging emphasized the fluidity of language use, where speakers drew from their entire linguistic repertoire to communicate effectively. This shift in perspective led to the expansion of translanguaging beyond Welsh-English contexts, influencing multilingual classrooms globally (Jaspers, 2018). In the late 1990s and early 2000s, translanguaging was further conceptualized by researchers such as García (2009) and Otheguy, García, and Reid (2015), who developed it into a robust sociolinguistic theory. This theory positioned translanguaging as a practice that not only facilitates communication but also serves as a form of resistance to monolingual ideologies, which often marginalize multilingual speakers. By recognizing the legitimacy of all linguistic resources, translanguaging has become a powerful pedagogical tool in advocating for language rights and equity in education. Furthermore, scholars like Canagarajah (2011, 2022) expanded translanguaging as an innovative teaching strategy, wherein both students and teachers strategically use multiple languages to foster deeper learning, encourage critical thinking, and bridge gaps between languages and cultures. This pedagogical shift reflects a growing awareness of the sociocultural and cognitive benefits of integrating multiple languages in the classroom, rather than enforcing strict language boundaries.

3. 

The reviewer stated that p. 7: Citations needed for language anxiety is a significant barrier to language acquisition.

I added related citations for the mentioned sentence. 

Language anxiety is a significant barrier to language acquisition (Aichhorn & Puck, 2017; Dryden et al., 2021; Horwitz, 2001, 2010; Wang, 2022).

4.

The reviewer asked p. 11-12: was the questionnaire pilot tested?

Yes, the questionnaire was pilot tested and related clarification was added into the study as follows:

Prior to its full implementation, the questionnaire was pilot tested with a sample of 30 pre-service EFL teachers to ensure clarity, reliability, and validity of the items. Based on the feedback from the pilot study, several items were revised for clarity, and a few ambiguous or redundant items were removed to improve the overall coherence and effectiveness of the instrument.

Reviewer #2: 

Reviewer #2: This is a clear and well written study of the use of translanguaging in the Turkish context. I have only a few suggestions for the authors:

1. I would love a few more details about the qualitative coding, specifically whether any inter-rater reliability was attempted or attained. 

The details regarding the qualitative coding process, including the assessment of inter-rater reliability, which was achieved using Cohen’s Kappa with a value of 0.82, indicating substantial agreement between the coders was added in the text as shown below.

To enhance the reliability of the coding process, inter-rater reliability was assessed by having two independent researchers code a subset of the interviews. The level of agreement between the coders was calculated using Cohen’s Kappa, which resulted in a value of 0.82, indicating a substantial level of agreement and high inter-rater reliability.

2. This may be a personal preference, but I do not like the presentation of a list of isolated quotes for qualitative results - (pp. 23-26). I'd love some author input between the quotes to provide context/situational description.

To address the reviewer's concern about providing more context and author input between the quotes, I added brief explanations and summaries before and after each set of quotes to give them context. In this version, each set of quotes is contextualized with thematic summaries and insights, providing the "author input" that the reviewer requested while retaining all the original quotes.

Perception of Translanguaging Practice

Students expressed a variety of opinions on the usefulness of translanguaging in their learning. While some viewed it as a natural and helpful tool for comprehending complex topics, others found it either unnecessary or less effective for their learning preferences:

I find translanguaging a very natural and helpful part of my learning. It makes it easier to understand complex topics.

For me, translanguaging is okay but only sometimes necessary. I can get by without it.

I do not see much benefit in using different languages in class. I prefer sticking to English.

This range of perspectives highlights the diverse ways in which students perceive the value of translanguaging, with some viewing it as integral, while others find it dispensable.

Personal Use of Translanguaging

When discussing their personal use of translanguaging, students varied in how frequently they relied on this practice. While some used it regularly to deepen their understanding, others preferred sticking to English unless they encountered a specific challenge:

I regularly use translanguaging in my studies to make sure I better grasp the meaning of new concepts.

I use it only when I am stuck. Otherwise, I try to stick to English.

I prefer to use only English in my classes, as I feel it is more effective.

These reflections show that translanguaging is a personalized strategy, with some students making it a central part of their learning, while others reserve it for specific circumstances.

Intentionality of Translanguaging

The intentionality behind translanguaging use was another key theme. For some students, it is a conscious strategy to manage anxiety and improve confidence, while for others, it occurs more spontaneously without deliberate intention:

I intentionally use translanguaging to ease my anxiety and boost my confidence in speaking English.

Sometimes I use translanguaging without thinking about it, but other times I do it intentionally to help with my learning.

I use different languages unconsciously whenever it happens during my learning process.

This mix of intentional and unintentional use underscores the varying levels of awareness students bring to their use of translanguaging as a learning aid.

Situations for Translanguaging

Students also identified particular situations in which they were more likely to engage in translanguaging. These tended to involve challenging aspects of language learning, such as complex grammar rules or unfamiliar vocabulary:

I mostly use translanguaging when dealing with difficult grammar rules or unfamiliar vocabulary.

I only use translanguaging if I have to, usually in particular situations.

I rarely use translanguaging unless necessary for understanding something.

These responses suggest that translanguaging is often reserved for moments when students face difficulties in their language learning process, highlighting its role as a support mechanism.

Classroom Environment Contributions

Students' opinions on the effect of translanguaging on the classroom environment were mixed. Some found that it improved participation and comfort, while others did not notice a significant difference or felt that it could detract from the focus of the class:

Translanguaging has improved my class participation and made me feel more comfortable expressing myself.

I do not notice much difference in the classroom environment when translanguaging is used.

In my experience, translanguaging sometimes makes the classroom environment less focused.

These contrasting views indicate that while some students perceive translanguaging as fostering a more inclusive classroom atmosphere, others feel that it does not have a meaningful impact on the overall class dynamics.

Language Acquisition Contributions

Regarding language acquisition, some students credited translanguaging with helping them understand English more effectively, while others felt its contributions were minimal:

Using translanguaging has greatly helped my understanding of English and made me feel more confident.

Translanguaging helps with my language learning, but not as much as I had hoped.

I do not feel that translanguaging contributes much to my language acquisition.

These differing viewpoints suggest that while translanguaging can be a valuable tool for some learners, its perceived effectiveness in language acquisition varies significantly.

Possible Disadvantages of Translanguaging

Some participants raised concerns about potential downsides to translanguaging, such as the risk of over-reliance on multiple languages or the possibility of confusing them. However, others saw no real disadvantages:

I worry that relying on translanguaging too much might weaken my English skills over time.

There are a few disadvantages, like potentially confusing the languages or relying on them too much.

I do not see any real disadvantages in using translanguaging.

These responses highlight both the perceived risks and the lack of concern for potential negative effects, underscoring that students' experiences with translanguaging are diverse.

Use of Different Languages in English Classes

The use of multiple languages in English classes elicited a range of opinions from students. Some found it enriching and inclusive, while others felt neutral or questioned its benefits:

Incorporating different languages in our English classes has made the environment richer and more inclusive.

I feel neutral about using other languages in class; it does not make much difference to me.

Using different languages does not seem to benefit my learning experience in English

These mixed responses suggest that the effectiveness of using multiple languages in English classes is highly context-dependent and varies from learner to learner.

Classroom and Cultural Contributions of Multilingualism

Many students acknowledged the broader cultural contributions of multilingualism, particularly in helping them understand and respect different cultures. However, the perceived impact of multilingualism was not universally felt:

Multilingualism has significantly helped me understand and respect different cultures better.

There are some benefits to multilingualism, but they are not as impactful as I expected.

I do not see how multilingualism contributes much to my understanding of other cultures.

This variation in perceptions indicates that while multilingualism can enhance cultural understanding for some, its benefits are not universally recognized by all learners.

Impact on Promoting Multiculturalism and Multilingualism

Finally, students reflected on the impact of translanguaging in promoting multiculturalism and multilingualism. While some saw clear benefits, others were less convinced of its impact:

Translanguaging has played a major role in improving my communication skills and understanding of different cultures.

It has some impact on multiculturalism and multilingualism, but not as much as I would have liked.

I have not noticed much of an impact on promoting multiculturalism or multilingualism from translanguaging.

These responses show that while translanguaging is viewed as a potential tool for fostering multicultural and multilingual competencies, its perceived influence varies among students.

3. I think the first paragraph of the discussion could benefit from paragraphs to organize the various hypotheses presented - having them all in one paragraph was a bit overwhelming. Because there are several hypotheses presented for the gender differences, I think it would be more readable to organize them in this way together.

I reorganized the discussion part as seen below. 

Discussion

This study explored pre-service EFL teachers' perceptions of, and attitudes towards, translanguaging strategies, their reported usage of these strategies in various instructional contexts, and the perceived benefits of translanguaging for English learners. The results provide valuable insights into how translanguaging is perceived and utilized in language education, while also comparing these findings with existing literature.

The analysis revealed gender differences in perceptions of translanguaging, with female participants generally holding more favorable views than their male counterparts across the scale and all its subscales. Several factors may contribute to these differences. First, research suggests that women often adopt more collaborative and inclusive communication styles (Peck, 2016; Wood & Inman, 1993), which might align with translanguaging practices promoting the use of multiple languages for understanding and learning (Ticheloven et al., 2021). This could explain why female pre-service teachers might be more receptive to translanguaging.

Second, language anxiety and confidence levels vary by gender (Geçkin, 2020). Female participants may view translanguaging as a tool that reduces anxiety and increases confidence, as reflected in their more positive perceptions (Dryden et al., 2021). Third, differences in educational experiences and exposure to multilingual education might influence perceptions

---

## [Decision Letter · Decision Letter 1]

4 Dec 2024

Breaking Down Linguistic Barriers: The Radical Impact of Translanguaging on Pre-Service EFL Teachers' Perspectives in Turkey

PONE-D-24-31275R1

Dear Dr. Ulum,

We’re pleased to inform you that your manuscript has been judged scientifically suitable for publication and will be formally accepted for publication once it meets all outstanding technical requirements.

Kind regards,

Mary Diane Clark, PhD

Academic Editor

PLOS ONE

Additional Editor Comments (optional):

Thank you fro your careful changes based on the reviewer and my earlier comments. I think that the paper will contribute to the literature.

Reviewers' comments:

Reviewer's Responses to Questions

**Comments to the Author**

1. If the authors have adequately addressed your comments raised in a previous round of review and you feel that this manuscript is now acceptable for publication, you may indicate that here to bypass the “Comments to the Author” section, enter your conflict of interest statement in the “Confidential to Editor” section, and submit your "Accept" recommendation.

Reviewer #1: All comments have been addressed

2. Is the manuscript technically sound, and do the data support the conclusions?

Reviewer #1: Yes

3. Has the statistical analysis been performed appropriately and rigorously? 

Reviewer #1: Yes

4. Have the authors made all data underlying the findings in their manuscript fully available?

Reviewer #1: Yes

5. Is the manuscript presented in an intelligible fashion and written in standard English?

Reviewer #1: Yes

6. Review Comments to the Author

Reviewer #1: I have no additional comments. The authors met all my recommendations for reviewers so I am satisfied for the response.

7. PLOS authors have the option to publish the peer review history of their article (what does this mean?). If published, this will include your full peer review and any attached files.

Reviewer #1: No

---

## [Editor Report · Acceptance letter]

11 Dec 2024

PONE-D-24-31275R1 

PLOS ONE

Dear Dr. Ulum, 

I'm pleased to inform you that your manuscript has been deemed suitable for publication in PLOS ONE. Congratulations! Your manuscript is now being handed over to our production team.

Kind regards, 

on behalf of

Dr. Mary Diane Clark 

Academic Editor

PLOS ONE